# Degradation of CdS Yellow and Orange Pigments: A Preventive Characterization of the Process through Pump–Probe, Reflectance, X-ray Diffraction, and Raman Spectroscopy

**DOI:** 10.3390/ma15165533

**Published:** 2022-08-11

**Authors:** Francesca Assunta Pisu, Pier Carlo Ricci, Stefania Porcu, Carlo Maria Carbonaro, Daniele Chiriu

**Affiliations:** Department of Physics, University of Cagliari, Cittadella Universitaria, 09042 Cagliari, Italy

**Keywords:** cadmium sulfide, degradation, Raman spectroscopy, transient absorption spectroscopy, reflectance, photoluminescence

## Abstract

Cadmium yellow degradation afflicts numerous paintings realized between the XIXth and XXth centuries. The degradation process and its kinetics is not completely understood. It consists of chalking, lightening, flaking, spalling, and, in its most deteriorated cases, the formation of a crust over the original yellow paint. In order to improve the comprehension of the process, mock-up samples of CdS in yellow and orange tonalities were studied by means of structural analysis and optical characterization, with the principal techniques used in the field of cultural heritage. Mock ups were artificially degraded with heat treatment and UV exposure. Relevant colorimetric variation appears in CIE Lab coordinates from reflectance spectra. XRD, SEM-EDS, and Raman spectroscopy revealed the formation of cadmium sulfate, whilst time-resolved photoluminescence and pump–probe transient absorption spectroscopy suggest the formation of a defective phase, compatible with Cd vacancies and the formation of both CdO and CdSO_4_ superficial clusters.

## 1. Introduction

During the nineteenth century, new inorganic pigments were synthesized and used extensively by coeval artists [1,2] for their characteristics, such as high color intensity, low cost, and covering power; they substituted the well-known ancient pigments in numerous Impressionist and early Modernist paintings.

All the pigments, organic and inorganic, synthetic and mineral, can be subjected to time corruption. Degradation can arise from environmental conditions, light exposure, bacterial attacks, or the use of erroneous solvents during the restoration process.

As is known, the most famous pigments used in painting are usually semiconductors. Depending on the surrounding environment, these materials are potentially not stable and degradation processes can take place through reactions with the atmosphere, especially with molecules, such as carbon dioxide, sulfur, oxygen, water, etc. [3,4]. The result implies the formation of patinas of new compounds due to carbonation or oxidation processes. Sometimes, the environmental conditions cause a phase transformation in the mineral that can present a different color. The reaction depends on many important parameters, such as temperature, light, relative humidity, pristine defect density in the materials, points of nucleation, time, etc.

We decided to concentrate our attention on nucleation processes due to an initial presence of defects in the structure. Those point defects could be related to different phenomena, such as, for example, light exposure. Once the degradation process is observed, one can trace back the numerous steps that led to the deteriorated materials, down to the presence of nucleation areas where the process started. Sometimes, these nucleation areas can be readily observed on the surface of the applied pigment. The questions we would like to answer are the following: Are we able to detect a primordial state of nucleation area where only starting point defects are present? Which non-destructive diagnostic tools can be promoted to detect those defects and to prevent degradation? The answer can arrive from the combined use of some optical technique, such as Raman spectroscopy, reflectance, and pump–probe spectroscopy [5,6,7,8,9,10]. An additional value can be associated to pump–probe spectroscopy, not conventionally used in the field of Cultural Heritage. Here, the electronic properties can be studied by analyzing all the optical transitions that involve not only the natural energy level positions, but even the presence of a distribution of some electronic levels due to color centers. These centers are associated with point defects, from which the mentioned nucleation process can start. In this work, we explored the combination of some conventional techniques and the pump probe one to provide an answer to the degradation of yellow cadmium pigment.

### 1.1. Yellow and Orange Cadmium Pigments

Yellow cadmium pigment is constituted by a cadmium sulfide (CdS) semiconductor, presenting both crystalline and amorphous forms. The hexagonal phase (α-CdS) is found in nature as the mineral greenockite, with space group P6_3_mc, the cubic structure belongs to the space group F$\bar{4}$3m (β-CdS), known as hawleyite mineral [11], while the amorphous form is a chemically synthesized product. 

The synthesis procedure can be divided into wet and dry processes. The latter could have, as starting materials, cadmium oxide, cadmium metal, or cadmium carbonate, each of which can be mixed with sulfur and heated to 300°–500° in the absence of air. The wet process consists in a solution with soluble sulfide (such as hydrogen sulfide, sodium sulfide, or barium sulfide) and soluble cadmium salts, such as chloride, cadmium nitrate, cadmium sulfate, or cadmium iodide, with a final precipitation of CdS [12,13]. 

During the 20th century, to produce different hues from the light yellow typical of CdS, its synthesis started to be changed by inserting zinc to lighten the color and selenium to increase the red hue. The use of selenium during the production allows one to obtain a range of colors from pale yellow to red, depending on the percentage of selenium, and these pigments are known as cadmium sulfoselenide compounds. Around 1920, to obtain another lighter shade, the use of additive barium sulfate was included in the synthesis procedure. These pigments can be prepared by calcining the hexagonal CdS plus selenium again or calcinating the precipitate resulting from the treatment of a cadmium salt with an alkaline selenide and sulfide [14].

### 1.2. CdS Degradation Phenomena

The degradation process leads to various alteration degrees in the original yellow paint till the formation of an overlaying crust in the worst cases [15]. Degradation of cadmium yellow was observed in works of many famous artists, such as Pablo Picasso, Vincent Van Gogh, Georges Seurat, Henri Matisse, Ferdinand Leger, Edvard Munch, and James Ensor [13,15,16,17,18].

Many studies were performed on Impressionist paints to understand the degradation process and identify which degradation products arise. In degraded areas of James Ensor’s paint, Van der Snickt et al. [13] found the formation of disfiguring white crystals that are present on the paint surface in approximately 50 μm sized globular agglomerations. They noted that the degradation penetrated below the paint surface about ca. 1–2 µm, suggesting that the degradation process appears to have affected only the material that is in direct contact with moisture and (UV) light. The degradation process is due to oxidation in the CdS pigment, with the final formation of cadmium sulfate as a degradation product. They found traces of CdCO_3_ and PbSO_4_. The former was attributed to a reaction from the starting material; the presence of the latter was not clarified. In the work of Van Der Snickt et al. [17], the authors suggested the presence of cadmium carbonate as a secondary degradation product following the primary photo-degradation of CdS, perhaps by the capture of atmospheric CO_2_, or a tertiary process involving a further breakdown in cadmium oxalate into cadmium carbonate. Further, Pouyet and coworkers [16] reported that the degraded white crust of the paints showed a high presence of CdCO_3_ and traces of CdSO_4_·nH_2_O and CdC_2_O_4_. The presence of cadmium carbonate was explained both as a starting reagent of the wet process and as second degradation product of photo-alteration.

Other degradation products individuated in degraded paints are cadmium oxalate, cadmium sulphate, and cadmium oxide. All of them can be derived from starting materials or filler products in paints but, as asserted by Mass et al. [15], their presence on the surface and not in the depth of the paint layer, is the proof of their degradation character. The cadmium oxalates are concentrated near the surface of the paint layer, demonstrating that this is a photo-oxidation product rather than a paint filler in the cadmium yellow paints. Cadmium sulfates have high solubility, so their presence in traces throughout the paint layer can be explained as mobile photo-oxidation products or, again, as starting reagents.

In addition to paintings, hexagonal CdS is one of the most important semiconductors for high-tech applications, for solar energy harvesting, among others, because of its bandgap around 2.4 eV [19] and calcination studies of synthetic CdS and consequent photo-oxidation by visible light exposure were carried out for this research field [20]. CdO was detected as a photo-degradation product and it was also found that the calcination treatment decreases the trap states. In addition, by increasing the calcination time, the color of the sample changes from yellow to pale yellow, with variations in the absorption curve that become sharper.

Therefore, the different degradation pathways of CdS pigment proposed in the literature can be summarized. The action of light with energy equal to or higher than CdS band gap leads to the formation of electron-hole pairs:(1)$$CdS+h\nu \to \;e^{-}+h^{+}$$

The holes oxidize the cadmium compound with:(2)$$CdS+2h^{+}\to Cd^{2+}+S.$$

At this point, the contact with air oxygen leads to the oxidation of sulfur to sulfate: (3)$$CdS+2O_{2}\to Cd^{2+}+SO_{4}^{2-}$$

More generally, the paint’s exposure to moisture and air can lead to a fading in yellow with the formation of cadmium sulfate hydrate, as confirmed by XRD measurements in [13]. Therefore, the chemical reaction is:(4)$$CdS+2O_{2}+H_{2}O\to CdSO_{4}\cdot H_{2}O$$

L. Monico et al. proposed another mechanism of sulfate formation considering the action of surface holes [21]:(5)$$CdS+4h_{surf}^{+}+2H_{2}O+O_{2}\to CdSO_{4}+4H^{+}$$

The presence of lead carbonate (not clear yet) with the formation of cadmium sulfate could cause the presence of lead sulfate as another degradation product:(6)$$PbCO_{3}+SO_{4}^{2-}\to PbSO_{4}+CO_{3}^{2-}$$

The degradation phenomena of CdS pigments are clearly related to changes in CdS optical proprieties. Comelli et al. [18], by studying the degraded paint of Picasso’s Femme, demonstrated that PL emissions from trap states (TS) are highly favored in relation to the near band edge (NBE) recombination, indicating a higher surface reactivity. This TS emission occurs at wavelengths shorter than those of not degraded CdS yellow. This behavior could be related to environment reactive nanometric-size grains of yellow pigment, resulting after the synthesis procedure without further calcination. In addition, a few computation studies in the literature [22,23,24] tried to explain the CdS degradation process and its origin.

In this work, to understand the degradation process of CdS pigment, the effect of calcination, the role of inner defects, and to clarify which degradation products could be originated, we simulated several artificial aging processes on two kinds of CdS commercial samples: the yellow and orange ones. The pigments were artificially degraded through different accelerated ageing processes: heat treatment and UV light exposure.

We studied the degradation phenomenon with various optical techniques in order to select which guarantees the fast results useful to determine the conservation state. In the past, artificial aging was already realized [12,21,25], aiming at studying the complex system made by the pigment with its binder (oil). In this study, we want to focus on the role of the pigment alone in the degradation process of CdS paints, after two different simulated artificial degradation processes, to understand which final degradation products are formed after them. In future work, the study will be extended to other components in the complex system of paint to include both the binder and the support.

## 2. Materials and Methods

### 2.1. Mock-Up Samples

Cd pigments used for accelerated degradation studies were bought from Kremer pigments. We used pigments number 21,040 (white yellow) and 21,080 (orange) and called them C-A and S-A respectively. The artificially aged samples were named with the initial C or S to indicate the yellow or orange paints, followed by a number indicating the temperature of heat treatment (300–400–500) and duration of heating (1 h, 2 h, 6 h). With the “UV” term we expressed the UV exposure realized by Hg lamp, with wavelength at 365 nm, followed by the relative exposure time (in hours) up to 56 h with a density power of 7 mW/ cm^2^. 

The thermal treatment is obtained using a temperature range between 300 and 500 °C for different heating times (1 h, 2 h, 4 h, 6 h). Their chromatic and structural variation were measured with micro-Raman spectroscopy, reflectance, TR-PL spectroscopy, X-ray diffraction, and transient absorption. Then these samples were stored at room temperature, 20% relative humidity (RH) in contact with air, illuminated by ambient light from Compact Fluorescent Lamps (CFL) for 4 h/days for 6 months. 

To perform UV exposure the acquired powder was mixed with distilled water and dispersed on a slide. The sample was exposed with Hg-lamp combined with a filter to remove the visible components and leave mainly the 365 nm component. The exposure was made at room temperature and with an RH value of 20%. The reflectance spectra were collected by step of 8 h a day and the other measurements are made only on the raw samples and at the end of the process for a total period of three weeks of air exposure. 

### 2.2. Reflectance Measurements

Reflectance measurements were performed by means of a Laser-driven Xenon lamp (EQ-99X) with wide emission spectrum from 200 to 2000 nm and average optical power of 1 mW/nm. The source was coupled with an optical fiber to an integrating sphere and to an Advantes Sensiline Avaspec-ULS-TEC Spectrometer (spectral range 250–900 nm). The measurements were acquired in 10° reflection mode, with a BaSO_4_ plate as reference. The results were related to the D65 illuminant and the CIELab standard colorimetric observables. The CIE coordinates were obtained using the ColorConvert v. 7.77 software.

### 2.3. Raman Measurements

High-resolution micro-Raman scattering measurements were obtained in back-scattering geometry through the confocal system SOL Confotec MR750 equipped with a Nikon Eclipse Ni microscope. Raman spectra were gathered by using, as excitation wavelength, the 532 nm line of a solid-state laser (DPSS LASOS Instruments, Jena, Germany). A grating with 1800 grooves/mm was used to obtain a resolution of 0.2 cm^−1^. Near-infrared micro-Raman scattering measurements were carried out in back-scattering geometry with a 1064 nm line of an Nd:YAG laser. Measurements were performed with a compact spectrometer B&WTEK (Newark, NJ, USA) i-Raman Ex integrated system with a spectral resolution of 8 cm^−1^. For each experimental setup, all the spectra were collected with an acquisition time of about 60 s (five replicas) and power excitation between 5 and 10 mW concentrated in a spot of 0.3 mm^2^ on the surface through a Raman Video MicroSampling System (Nikon, Tokyo, Japan) Eclipse for high-resolution and BAC151B in the other case equipped with a 20× Olympus objective to select the area on the samples. Each measurement area is represented by 5 measures over the surface of 1 cm^2^ and the average value of all the spectra is proposed in the experimental results. A total variation of less than 2% assures homogeneity in the surface degradation.

### 2.4. Time-Resolved Photoluminescence (TR-PL)

TR-PL measurements were recorded by using different excitation systems:Excitation with 100 fs long pulses delivered by an optical parametric amplifier (Light Conversion TOPAS-800-fs-UV-1) pumped by a regenerative Ti:sapphire amplifier (Coherent Libra-F-1K-HE230). The repetition frequency was 1 kHz.Excitation with 100 fs long pulses from Ti:sapphire oscillator Coherent Chameleon Ultra II having a repetition rate of 80 MHz.

PL signal was recovered by a streak camera (Hamamatsu C10910, Hamamatsu, Photonics, Herrsching am Ammersee, Germany) equipped with a grating spectrometer (Princeton Instruments, Trenton, NJ, USA, Acton Spectra Pro SP-2300). All of the measurements were collected in the front-face configuration to reduce inner-filter effects. Proper optical filters were applied to remove the reflected contribution of the excitation light.

### 2.5. Pump and Probe Transient Absorption Spectroscopy (TAS)

Transient absorption measurements were performed with a pump–probe differential spectrometer (Ultrafast Systems HELIOS-EOS), with both pump and probe wavelengths generated by a Ti:Sapphire regenerative amplifier (Coherent Libra-F-1K-HE-230), which delivers 100 fs long pulses at 800 nm with 1 KHz repetition rate. The main emission from the regenerative amplifier was split into two branches: one sent to an optical parametric amplifier (Light Conversion TOPAS C), in order to generate the pump wavelengths, and the other sent to the sapphire plate of the HELIOS spectrometer, where multi-color probe beam was generated by means of white light supercontinuum generation. The probe pulses were time delayed with respect to the pump pulses, by passing through a variable digitally controlled optical delay line. The pump and probe beams were then non-collinearly focused and overlapped on the sample surface, with the pump being chopped at 500 Hz, so that half of the transmission spectra were recorded with the pump on and half with pump off. The transmission spectra from the probe beam were recorded as a function of the relative delay time, by means of CCD spectrometers

### 2.6. XRD Measurements

X-ray patterns were collected at room temperature by using a Bruker D8 Advance diffractometer operating at 30 kV and 20 mA, equipped with a Cu tube (λ = 1.5418 Å), a Vantec-1 PSD detector, and an Anton Parr HTK2000 high-temperature furnace. Powder patterns were recorded in the 15° ≤ 2θ ≤ 85° range.

### 2.7. SEM-EDS

SEM images were gathered by a scanning electron microscope ESEM: FEI Quanta 200 under low-vacuum conditions. EDS semiquantitative analyses were obtained with the help of Thermo Scientific (Waltham, MA, USA) EDS UltraDry INTX-10P-A system equipped with Pathfinder. Each point of analysis was collected with an acceleration voltage of 20 kV and live time of 30 s.

## 3. Results and Discussion

### 3.1. Heating Process

The starting pigments were characterized first by XRD to obtain information about the phase composition and additive compounds. At a later time, a second analysis with reflectance and Raman spectroscopy was conducted to have more detail on structural modifications during the aging process. XRD analyses revealed the presence of barium sulfate (see Figure 1a), in a percentage of about 7(1)% with respect to the remaining Cd_1-x_Zn_x_S in yellow cadmium (x = 0.19). The orange one revealed the presence of hexagonal CdS_1-x_Se_x_ and CdS (Figure 1b). The phase identification confirms the mixture of CdZnS and BaSO_4_ for yellow C-samples and CdSeS for orange S-samples. Thermal treatments on the two mentioned samples, C-A and S-A, were made at different temperatures (300 °C, 400 °C, 500 °C) for 1 h and, for the 500 °C temperature, we performed the calcination also for 2 h, 4 h, and 6 h until obtaining a notable chromatic change. The XRD measurements performed for the samples heated at 300–400–500 °C for 1 h are reported in Figure 1a,b. As displayed in the patterns, a perceptible difference is shown only in the C-500-1 h curve, in which a broadening of the region between 24 and 34° is present. This broadening could be associated with an increase in structural disorder related only to the CdZnS compound. Actually, studies in the literature on thermal stability on BaSO_4_ demonstrate that its XRD pattern did not change for this compound [26]. Contrary to what was discussed in the introduction, no evidence can be reported in this analysis about the formation of other compounds, such as CdSO_4_ and CdO. Only in the case of heat treatment at 500 °C for 6 h, the formation of these compounds is detected, justifying the observed change in color. Actually, by increasing the heating time until 6 h at 500 °C, the XRD measurements were able to detect the degradation products. As can be seen from Figure 1a,b, the C-500-6 h patterns show peaks of CdSO_4_·2CdO and in the S-500-6 h sample, also the presence of a pure Cd-sulfate phase.

The chromatic variation among the samples was measured from the reflectance spectra. The first derivative spectra for the yellow cadmium are shown in Figure 1c and the strongest variations recorded at 500 °C after 6 h are summed up in the CIE color chart in Figure 1d. The calculated CIE coordinates, relative simulated colors, and the maximum value of the first derivative (X_0_) are listed in Table 1.

By heating the C-A sample, a notable shift in X_0_ towards high wavelengths is produced. The shift in the S-sample is not remarkable, as can be seen from the X_0_ values in Table 1, where the peaks of the first derivative range from 514.5 and 556 (for the no-treated S-sample) to 515 and 556 (for S-500-6 h). As is known from the literature, the X_0_-value is linked to the direct bandgap value [27,28,29], so the shift in sample C is compatible with a decrease in the direct bandgap and it can be explained by the formation of the CdO compound after a thermal treatment, as proved by [20,30]. On the contrary, reflectance measurements suggest another change: heated samples reveal an increment in the luminosity L parameter in CIE Lab coordinates. This change could be associated with the formation of whitish compounds, probably CdSO_4_ [31].

As the calcination time increases, we generally register growth in the L parameter (Table 1) for both S-A and C-A samples. We can explain this behaviour considering that the colorimetric coordinates can be influenced by the combination of grain dimensions with the formation of mentioned compounds, CdO and CdSO_4_. If the temperature increases, the grain size in the sample is larger (see Comelli et al.) and this means a decrease in luminosity but also a major formation of cadmium sulfate and the presence of a darker compound, such as CdO. Therefore, the effect of the co-presence of a whitish and a darker compound, in addition to the variation in grain sizes of the sample, can influence the variability in the L parameter, which does not follow a linear increase as a function of the heat treatment. Actually, luminosity L decreases again after 6 h at 500 °C, during which, as hypothesized before, a predominant brownish compound is forming (CdO), inducing a reduction in this parameter.

To confirm this assumption, we performed TR-PL measurements and the results are included in Figure 2a: we reported the emission of C-samples before and after the thermal treatment, where, again, a visible red shift is present. Actually, the peak moves from 480 nm in sample C-A to 510 nm in the sample heated at 500 °C for 6 h. In Figure 2b, the TR-PL kinetics for the S-samples that do not present substantial variation as a function of the temperature are shown.

A Gaussian deconvolution of the spectra for different temperatures was calculated for sample C and the respective band positions in eV are represented in Table 2. TR-PL analysis was made on the time scale, ranging from a picoseconds regime to the nanosecond scale, revealing the presence of three decay times derived by three emission channels ($\tau_{1},\;\tau_{2}\;\mathrm{and}\;\tau_{3})$. In the case of the CA sample, as indicated by previous studies in the literature [32], the shorter time t_1_ around 8–13 ps is attributable to the bandgap emission, while t_2_ < 100 ps and t_3_ of 730 ps are associated with superficial and intermediate structural defects, respectively. While t_1_ and t_3_ times remain constants as a function of the temperature, the intermediate time t_2_ changes assuming a maximum value of 200 ps at 400 °C. This behavior seems to be compatible with the formation of CdO nuclei. Actually, in the literature, pure CdO presents two decay times, one in a range of 100–500 ps and one in a range of 1–3 ns. [33] This assumption is not exhaustive for the presence of this compound and a detailed study is necessary to corroborate this hypothesis.

TRPL was performed also in the microseconds range, confirming the presence of deep trap states (TS) emissions [32,34], with a slow broad emission in the spectral region 650–750 nm and a strong sharp peak at 790 nm due to crystal defects (see Figure 2c,d). Even using this technique, no evidence of time decays ascribable to CdSO_4_ can be reported, leaving unresolved the variation in the L paramater discussed before, associated with the formation of a withish compound.

To shed light on this behavior, a complete Raman characterization of the samples, at different wavelengths, was essential to better understand which probable compounds are formed by heating. For confirming the CdO formation hypothesis, the Raman spectra could not be helpful. In the literature, there are different and contrasting Raman spectra associated to cadmium oxide [35,36,37]; otherwise, according to some authors, cadmium oxide should not be Raman active [38]. Therefore, the individuation of this secondary compound cannot be undoubtedly approved from our Raman spectra.

However, the other discussed possible compound was detected. Actually, the Raman spectrum collected with 1064 nm shows the presence of a big amount of sulfate compound (Figure 3a), as demonstred by the presence of a strong peak around 1000 cm^−1^, typical for SO_4_ vibration modes. The existence of this band is revealed also in the samples treated at lower temperatures than 500 °C. To define, with greater precision, the position of the sulfate vibration and, therefore, to be able to identify the compound, the Raman spectra were acquired with a better resolution (1 cm^−1^), with a 532 nm source. The Raman spectra with high resolution presented a shift of about 6 cm^−1^ between the peaks due to the heat treatment, visible also with the help of a deconvolution procedure by Lorentzian curves around 990–1000 cm^−1^ (SO_4_ vibration), see Figure 3b, confirming the whitish CdSO_4_ formation.

As previously discussed, no peaks related to CdO and CdSO_4_ compounds at lower temperature than 500 °C are noticeable by XRD, suggesting the possible formation of a thin superficial layer of whitish CdSO_4_ crust, undetectable with the XRD technique because of the detection limit threshold. Only a broadening in the region between 24° and 34° was revealed, suggesting a progressive structural disorder in the phase CdZnS. This hypothesis can be confirmed with a detailed analysis of some vibrational modes of the Raman spectra. As reported in the literature [39,40] the ratio between the 215 cm^−1^ (TO multi-phonon process) and 300 cm^−1^ (LO) peaks of Cd-pigment Raman spectrum can provide information about the structural disorder, comparing the spectra of the natural and heated samples. In our case, the calculated ratio is drawn in Figure 3c. As reported in [40], the increase in structural disorder and zinc content leads to a decrease of 215 cm^−1^ band for the TO mode. In our yellow samples, the heat treatment produces a decrement in TO-peak intensity, as can been seen directly by the spectra reported in Figure 3c, confirming the hypothesis.

Finally, to obtain further confirmation and information about the superficial effect of heating treatment, SEM-EDS measurements were also performed on these two samples. In Table 3, stoichiometric calculations on the element percentages to reach Cd saturation suggest, again, the presence of hydrate sulfate compounds and almost a double amount of Cd-oxide.

### 3.2. UV Exposure Process

In order to establish the light stability in Cd pigments used in paints, a UV treatment at 365 nm, with different exposure times, was conducted. To characterize the pigment variations, after this degradation process, we adopted Raman spectroscopy, reflectance, luminescence, and transient absorption. As found in the literature [21], the UV action in the presence of oxygen can produce the formation of sulfate compounds.

In particular, Raman analyses made on the yellow sample confirmed a broad band in the region of sulfate vibrations between 990 cm^−1^ and 1008 cm^−1^ (see Figure 4a).

To establish if the new sulfate compound is derived from CdS degradation or if the starting barium sulfate converted to another form, in Figure 4b, the deconvolutions (Lorentzian functions) for the artificially and no-degraded samples are presented in the region of interest (R.O.I). The C-UV-56 h signal consists of two peaks, one located at 994 cm^−1^ with an area of 0.89 and the other one at 1007 cm^−1^ with an area of 0.5. The former is very similar to the one of the C-A band (grey line) located at 992 cm^−1^ (attributed to BaSO_4_). The deconvolution procedure presents a slight red shifting and about a doubling of the relative area, suggesting a conversion of barium sulfate to another form. The latter can be associated with CdS degradation in cadmium sulfate. In fact, the peak at 1007 cm^−1^ is usually associated with cadmium sulfate compounds bound with the $x\cdot H_{2}O$ molecule [41]. To confirm the formation of this species, a compositional analysis by means of SEM-EDS was made and reported point by point in the Appendix A. For the C-UV-56 h sample, the results summarized in Table 4 confirm the presence of Cadmium hydrate sulfate and a possible excess of -S^2−^, suggesting the formation of a notable amount of Cd vacancies inside the CdS crystal after light exposure. This is another known cause for the color change in the Cd pigment, as previously reported in the literature [22]. Actually, the Cd vacancies led to the formation of an intra-gap level with NIR emission and time decays of some microseconds, as already discussed before in the case of thermally treated samples. 

With the intention of determining the real effect of color change, a detailed colorimetric analysis was performed. Reflectance spectra were collected with 8 h steps of UV exposure and the related CIE Lab coordinates were calculated. The results are summarized in the Appendix A. The L parameter tends to increase, mainly for the C-A sample with light exposure and the shape of the first derivative (Appendix A), after 56 h, widens, showing a blue shift, typical of the addition of light and white shades in yellow pigments, as studied by Gueli et al. [31]. The total variation in CIE coordinates is represented in CIE space. The S-A sample did not show a remarkable difference in CIE coordinates after 56 h of UV exposure as visible from the color space diagram (see Appendix A).

To speed up the reading of chromatic variations, a graph with the relative value for each coordinate is shown in Figure 4c, where we tried to express the amount of the total color variation by the ΔE value through a first kinetic model (exponential fit) for the C-A sample:(7)ΔE=A(1−e−tτ)
where *A* is the asymptotic value of the ΔE curve (the final step of conversion of CdS to CdSO_4_, involving the chromatic change from yellow colors to white ones) and tau is the characteristic time of reaction. After a time of about 40 h, the conversion is completed.

To understand what UV light accomplishes in the process, we monitored the stability of the studied pigments, kept in standard environmental conditions, and we observed that, effectively, the degradation also started slowly in a natural way. After a deposition above a slide, the samples, heated and no-heated (Figure 5), were kept at room temperature with 20 RH%, illuminated by artificial light (Compact Fluorescence Lamps) for 4 h/days for 6 months. NIR-Raman spectra were acquired showing the development of a new shoulder in the sulphate region. In Figure 5, the deconvolution of this region is represented. Even in this case, the formation of sulphate compounds was recorded, revealing that the action of light and mainly of the oxygen leads to degradation in Cd pigments.

The optical variations registered on cadmium yellow can be explained in detail with an in-depth study of the electronic properties of this pigment and its behavior after the accelerated degradation process. For this reason, we characterized, with pump–probe spectroscopy, the optical differences induced in our samples after UV exposure and heat treatment. To clarify the used nomenclature, we will describe, as a short-lived signal, those that last some picoseconds up to tens of picoseconds and long-lived signals, those ranging from hundreds of ps to ns. As reported in Figure 6a, the C-A sample shows a broad positive signal (excited state absorption—ESA), centered at 477 nm, with a duration of 300 ps. In addition, a shorter negative broad signal in the region between 650 nm and up to 800 nm is presented.

As a comparison, thermal treatment at 500 °C for 6 h drastically changed the previous ESA signal (Figure 6b). In this case, the spectrum is composed mainly of a broad short-lived signal, having different peaks in the region between 475 and 580 nm. Before 475 nm, the same ESA observed for the C-A sample is observed, but with a duration of only 5 ps. The broad short-lived signal after 20 ps converted into a negative signal, probably stimulated emission (SE). Around 750 nm, we have another positive signal with the same duration (about 20 ps) but with a longer rise time. The ground state depletion (GSD) signal in the near infrared is located only in the region 775–815 nm and with a very short time of about 5 ps. The disappearing of the broad trap state band (negative signal between 600 and 750) after the thermal treatment agrees with the TR-PL measurements in the micro-second scale mentioned before (Figure 2c).

The UV exposure led a further broadening of the ESA signals (Figure 6c) with respect to the C-A sample until 630 nm, after which a broad short-lived negative signal started from 640 nm. Additional positive bands start a few picoseconds after the pump absorption, suggesting a non-radiative relaxation to a lower level, from which starts the probe absorption.

Concerning the orange sample, the no-treated powder is composed of a long-lived ESA signal centered at around 495 nm. No negative signals were detected for this sample (Figure 6d). The heated sample (Figure 6e) showed the same ESA signal and, in addition, a new very broad long-lived negative signal in a region between 550 nm and 815 nm (2.25–1.52 eV) attributed to a GSD. This result is compatible with TR-PL measurements where the new trap states around 750–820 nm were obtained after the heating relax after microseconds (Figure 2d).

The UV-exposed sample showed variations in the kinetics of ESA signal, which became shorter (duration of about 40 ps) and the formation of a new broad negative signal centered at 570 nm. The latter can be broken up into two regions with different kinetics, as can be clearly seen from Figure 6f, associable to a long-lived GSD signal at 570 nm and short-lived GSD at 650 nm, both towards trap states.

To produce a first interpretation of these results, we can hypothesize that the negative signal in the NIR around 780 nm, found in both normal and UV-exposed samples, could be linked to Cd vacancies, as confirmed by SEM-EDS calculation and by previous authors [22]. Furthermore, the substantial difference in UV exposure for the C sample resides in the broadening of absorption signals, attributable to a change inside the conduction band structure, new levels due to the formation of defects, and changes in electronic transfer, as suggested by different kinetics observed in some positive signals. The thermal treatment and UV exposure, mainly for orange samples but even, to a lesser degree, in white yellow, led the formation of new trap states inside the bandgap, defects responsible for the darkening effect in aged samples.

## 4. Conclusions

In summary, in this study, we thoroughly analyzed the degradation of CdS yellow pigment used in paintings, with the purpose of shedding light about the possible causes and providing useful information for the field of Cultural Heritage. Light-yellow and orange pigments from Kremer were artificially degraded through different accelerated ageing: heat treatment and UV light exposure. Whereas the orange pigment seems more stable, the yellow one degrades more markedly. Reflectance spectra and chromatic coordinates in the CIELab space revealed that the heating treatment, executed in a range of 100–500 °C until 6 h, causes a prominent color variation for the light-yellow pigment in terms of bleaching. XRD and Raman spectroscopy suggest that the cause is attributable to the formation of sulfate compounds and possible cadmium oxide, detected in the XRD pattern of the sample heated at 500 °C for 6 h, after the reaction with the oxygen present in the environment atmosphere. This interpretation is also confirmed by the punctual SEM/EDS analysis. The formation of deep trap states and oxide products after thermal treatment was also confirmed by TRPL measurement, performed in the ps and micro-second scale. The same conclusion for the formation of whitish compounds can be adopted for UV treatment, which allows one to demonstrate, by means of Raman spectroscopy and SEM-EDX, the formation of a superficial sulfate phase. In addition, the action of UV light for the yellow sample, and both UV and thermal exposure for the orange one, seems to produce a defective phase where intra-gap energy levels are generated. Actually, with the help of pump–probe measurements, GSD and long ESA signals due to the formation of trap levels are evidenced in the visible and near-infrared region, both for light-yellow and orange pigments. If structural defectivities are activated by light exposure, the reaction with atmosphere seems to also produce darker compounds, such as cadmium oxide, and whitish compounds, such as cadmium sulfate, as previously reported in the literature.

## Figures and Tables

**Figure 1 materials-15-05533-f001:**
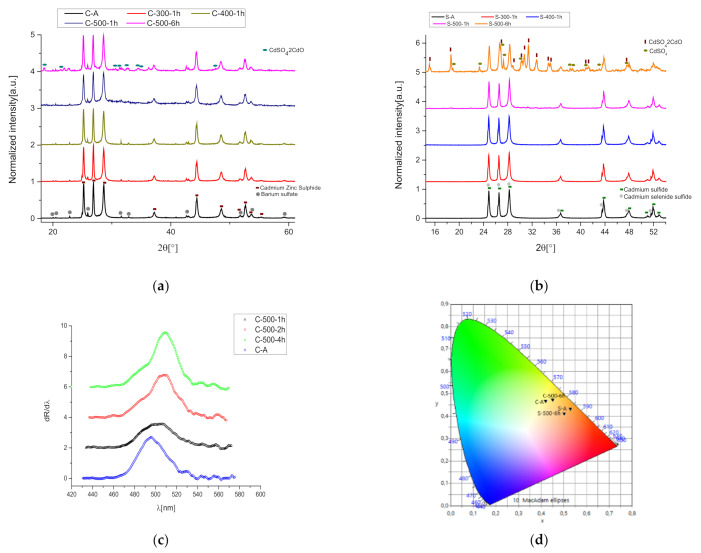
(**a**) XRD patterns of C-A heated and no-aged samples; (**b**) XRD patterns of S-A heated and no-aged samples; (**c**) first derivative reflectance spectra for heated C-A samples; and (**d**) the color chart of C-A, S-A, C-500-6 h, and S-500-6 h samples.

**Figure 2 materials-15-05533-f002:**
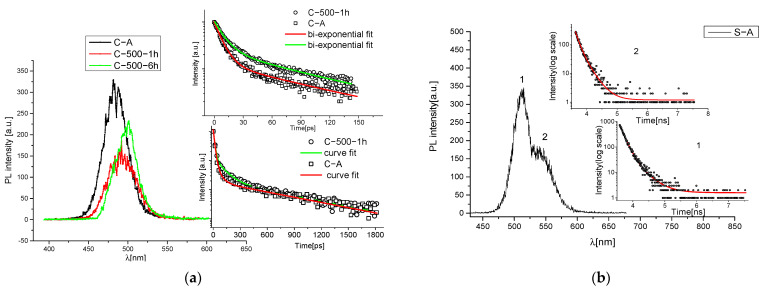
(**a**) Comparison between the PL emission of C-A sample and the C-samples heated at 500 °C for 1 h and for 6 h and respective decay fits for C-A and C-500-1 h samples; (**b**) TRPL spectra of S-A sample; in inset, the time-resolved spectra of the two bands of emission (one related to CdS and the second to -Se inclusion) with relative bi-exponential decay fit; (**c**) TRPL microsecond-scale analysis for C-A and C-400-1 h samples; in inset, time-resolved fit of 700–800 nm emission band for both the samples; (**d**) TRPL microsecond-scale analysis for S-A and S-400-1 h samples; in inset, time-resolved fit of 700–800 nm emission band, exciting wavelength 430 nm, P = 50 uW.

**Figure 3 materials-15-05533-f003:**
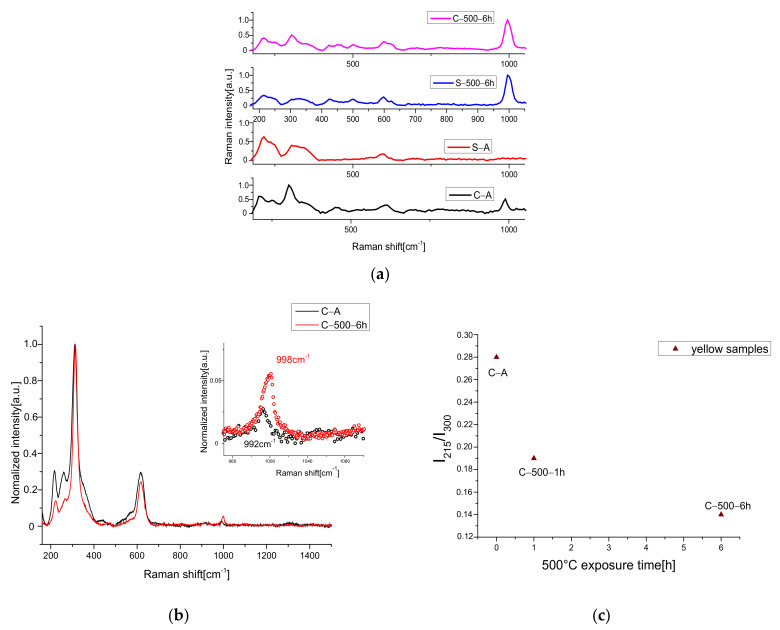
(**a**) 1064 nm excited Raman spectra before and after 500 °C for 6 h; (**b**) 532 nm-Raman spectra of C-A and C-500-6 h; in inset, the R.O.I with the value of the peaks obtained by the deconvolution process. (**c**) Variation in TO/LO bands in high-resolution 532 nm excited Raman spectra of yellow samples.

**Figure 4 materials-15-05533-f004:**
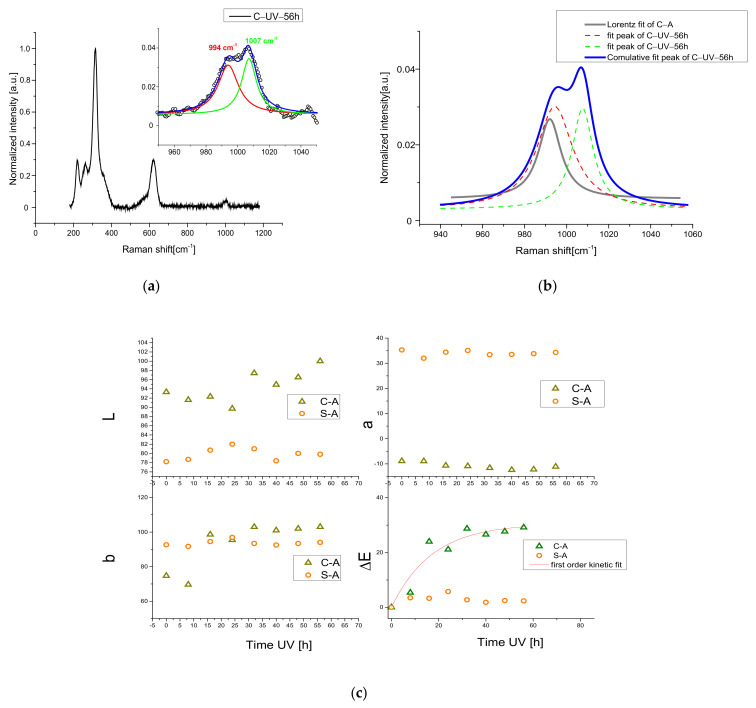
(**a**) Raman spectrum of C-UV-56 h sample with 532 nm excitation wavelength; in inset, the deconvolution of the region of sulfate; (**b**) deconvolution of the region 900–1060 cm^−1^ of C-A Raman spectrum; (**c**) CIE coordinate trend for C-A and S-A samples exposed to UV radiation. The ΔE for the yellow sample shows exponential growth.

**Figure 5 materials-15-05533-f005:**
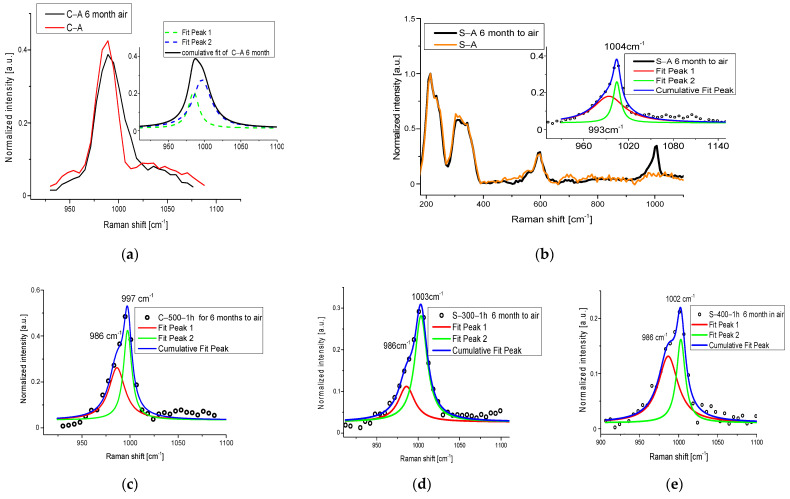
(**a**) Raman spectrum of C-A sample exposed to 6 months in air in a range between 900 and 1110 cm^−1^; in inset, the band deconvolutions; (**b**) Raman spectra of S-A sample and S-A after 6 months in air; in inset, the deconvolutions of the R.O.I (sulphate region) for the sample exposed to air; (**c**) deconvolutions of the R.O.I for the C-500-1 h sample exposed to 6 months to air; (**d**) deconvolutions of the R.O.I for the S-300-1 h sample exposed for 6 months to air; (**e**) deconvolutions of the R.O.I for the S-400-1 h sample exposed for 6 months to air.

**Figure 6 materials-15-05533-f006:**
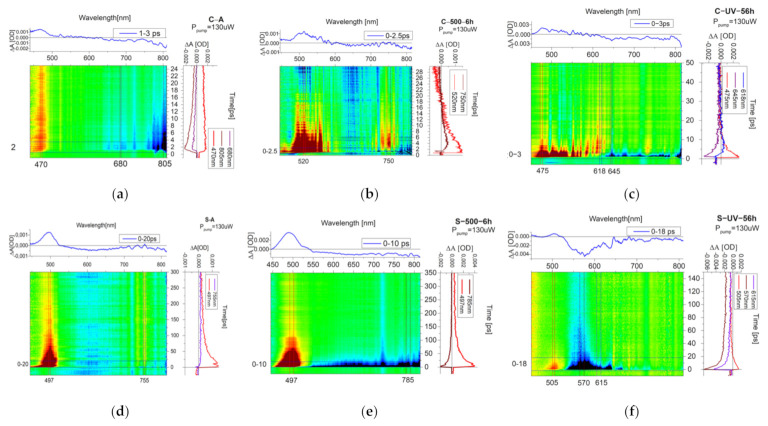
Transient absorption map of (**a**) C-A sample, (**b**) C-500-6 h sample, (c) C-UV-56 h sample, (**d**) S-A sample, (**e**) S-500-6 h sample, and (**f**) S-UV-56 h sample.

**Table 1 materials-15-05533-t001:** Chromatic variation in C-A and S-A samples after heating process obtained by reflectance spectra.

Sample	L	a	b	X_0_ [nm]	Simulated Color	Sample	L	a	b	X_0_ [nm]	Simulated Color
C-A	94.1	−11.1	101	496.9	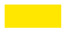	C-A	94.1	−11.1	101	496.9	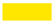
C-300 °C-1 h	103	−17.3	101	494.2		C-500 °C-2 h	100	−5.91	84	506.4	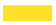
C-400 °C-1 h	93.7	−8.89	101	497.7	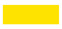	C-500 °C-4 h	104	−5.66	99.3	508	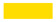
C-500 °C-1 h	85.8	−5.08	75.6	502.3	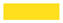	C-500 °C-6 h	91	−0.81	86.9	511	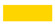
S-A	78.2	35.3	92.6	514.5, 556.0	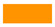	S-A	78.2	35.3	92.6	514.5, 556.0	
S-300 °C-1 h	92.2	42.4	107	512.5	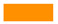	S-500 °C-2 h	89,7	37.1	95	514.0, 553.0	
S-400 °C-1 h	76.8	28.4	85.3	514.0, 552.8	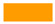	S-500 °C-4 h	91,5	43.2	105	513.0, 555.0	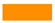
S-500 °C-1 h	76.4	29.9	89.7	514.3, 552.9	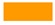	S-500 °C-6 h	70.4	32.1	62.6	515.0, 556.0	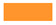

**Table 2 materials-15-05533-t002:** Value of emission channels for exciting wavelength λ = 450 nm, P = 235 μW.

Sample	E (eV)	*τ*_1_ [ps]	*τ*_2_ [ps]	*τ*_3_ [ps]
C-A	2.56	8–13	50	730
C-300-1 h	2.55	8–13	150	727
C-400-1 h	2.56	8–13	200	730
C-500-1 h	2.51	8–13	100	728
C-500-6 h	2.48	8–13	50	707

**Table 3 materials-15-05533-t003:** Calculated Cd- compounds percentage by saturation of Cd- amount for S-500-6 h and C-500-6 h.

Calculation of Cadmium Saturation for S-500-6 h Sample		
	**Point 1 [%]**	**Point 5 [%]**
Cd (Se,S)	26.3	27.6
CdSO_4_(8/3 H_2_O)	24.4	26.3
CdO	49.3	46.1
	**Point 2 [%]**	**Point 3 [%]**
CdZnS	32.7	38.7
CdSO_4_(8/3 H_2_O)	32.2	23.7
CdO	35.1	37.6

**Table 4 materials-15-05533-t004:** Calculated Cd- compound percentage by saturation of Cd- amount.

C-UV-56 h	Point 1 [%]	Point 2 [%]
CdZnS	13.3	64
CdSO_4_ (8/3 H_2_O)	86.7	36
CdO	-	-
-S^2−^ in excess	1.3	1.7

## Data Availability

Not applicable.

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
