# Peer review of "Degradation of CdS Yellow and Orange Pigments: A Preventive Characterization of the Process through Pump–Probe, Reflectance, X-ray Diffraction, and Raman Spectroscopy"

_materials, 2022, doi:10.3390/ma15165533_

Round 1

Reviewer 1 Report

The manuscript by D. Chiriu et al. presents a systematic study of degradation process of CdS yellow pigment under heat treatment and UV light exposure. In the study, the authors employed various characterization techniques to verify the degradation products. Overall, this manuscript meets the quality of Journal of Materials and I recommend its publication. I do not have too many comments about it. There are several layout problems as well as typos format and layout in the manuscript. For instance, the captions or legend of figure 2, figure 4 and figure 6 are not clear or incorrect. The format of legends in some figure is not consistent. These mistakes could mislead readers. The authors should pay more attention on these and go through the manuscript carefully to avoid such typos.

Author Response

We thank the reviewer for the kind comments. We apologize for the mistakes reported and we operated all the corrections required. In yellow are all the changes done in the manuscript. 

Reviewer 2 Report

In this work, authors studied mock-up samples of CdS in yellow and orange tonalities using various structural and optical characterization techniques. The samples were artificially degraded with UV exposure and heat treatment. Apart from XRD, SEM-EDS, and Raman spectroscopy authors also used transient pump probe spectroscopy to gain insights on the degradation of CsD pigment. Authors constructively presented their results and performed substantial work and the work is of good interest for the researchers developing spectroscopy techniques for cultural heritage. There are only a few to none grammatical mistakes, according to me, hence the article looks good can be considered to publish in the present form with few minor revisions.

1.     Line 1: During the nineteenth century, new inorganic pigments were synthesized and used extensively by coeval artists. Provide few examples found in literature.

2.     Line 66: Does this paragraph describes the chemical synthesis process of amorphous form? If yes, please mention it as “The synthesis procedure of amorphous CdS can be divided…”

3.     Line 87: Is it CdCO3 or CaCO3?

4.     Line 132: “trap s (TS)”to be written as “trap states”.

5.     Line 135-136: “calcination lack after synthesis procedure” What is the lac in the senstence refers to?

6.     Line 141: Why couldn’t authors test the amorphous CdS degradation? Could authors supplement information regarding the limitations on their sample set.

7.     Line 183: Several instances authors used, as an example in line 183 “were obtained by using the ColorConvert” authors used by using. Authors should use either by or using. This would be more appropriate if to write “were obtained using the ColorConvert””. Change it in the other instances as well.

8.     Line 199: “Each measurement area represents a sampling surface of about 1 cm2”. How many locations were used to obtain the Raman spectra? Did authors performed Raman mapping to study the homogeneity of sample degradation? If not, this could be included in the MS, as it shows the homogeneity of sample degradation.

9.     Line 280: Figure 1A label to be changed form A-C to C-A.

10.  Line 285: Table 1: Why does the S-300°C-1h has high luminosity of 92.2? It is the highest reported values for S films.

11.  Line 288: Did authors observe similar shift in S-A samples after calcinations of various times at 500°C? supplement the data.

12.  Line 138-140: “It was decided”. Could authors share an explanation or cite a reference for relation between size of the NPs to oxidation rate/ ionic silver release.

Author Response

We thank the reviewer for this kind report. We read with attention the comments and we replied point by point to the different questions /corrections:

  1. Line 1: During the nineteenth century, new inorganic pigments were synthesized and used extensively by coeval artists. Provide few examples found in literature.

      Accordingly with the reviewer, we added references on this part.

  1. Line 66: Does this paragraph describes the chemical synthesis process of amorphous form? If yes, please mention it as “The synthesis procedure of amorphous CdS can be divided…”

We thank the reviewer, but the mentioned processes are not dedicated to amorphous CdS. The proposed syntheses are both related to the hexagonal phase, the favoured one for pigment production because of its greater stability.

  1. Line 87: Is it CdCO3or CaCO3?

It was CdCO3, we corrected it in the text.

  1. Line 132: “trap s (TS)”to be written as “trap states”.

We corrected it on the text.

  1. Line 135-136: “calcination lack after synthesis procedure” What is the lac in the senstence refers to?

As the reference authors  wrote “ a subset of CdS pigments produced in the late 19th to early 20thcenturies were left uncalcined after synthesis, a procedure that would lead to reactive nanocrystalline CdS grains”

During 20th the  CdS synthesis methods included also  a calcination procedure to “stabilize” the pigment crystallinity, the wrong procedure of calcination or the total absence of this step (we use lack in this sense)   could be related to a higher  degree of reactivity/degradation  with the environments.   We reformulated the sentence as follows: “This behavior could be related to environment reactive nanometric size grains of yellow pigment resulting after synthesis procedure without further calcination”

  1. Line 141: Why couldn’t authors test the amorphous CdS degradation? Could authors supplement information regarding the limitations on their sample set.

     We don’t have the amourphous CdS, the XRD confirm the hexagonal phase of our starting samples.

  1. Line 183: Several instances authors used, as an example in line 183 “were obtained by using the ColorConvert” authors used by using.Authors should use either by or using. This would be more appropriate if to write “were obtained using the ColorConvert””. Change it in the other instances as well.

      We thank the reviewer; we changed the whole text accordingly.

  1. Line 199: “Each measurement area represents a sampling surface of about 1 cm2”. How many locations were used to obtain the Raman spectra? Did authors performed Raman mapping to study the homogeneity of sample degradation? If not, this could be included in the MS, as it shows the homogeneity of sample degradation.

We thank the reviewer for the comment. Unfortunately, we did not have the possibility to perform Raman mapping in our samples. We performed 5 measures over the surface of 1 cm2 and we propose the average values of all the spectra that presented a total variation of less than 2%.  We specify in the text as follows (line 210): “Each measurement area is represented by 5 measures over the surface of 1 cm2 and the average value of all the spectra is proposed in the experimental results. A total point-to-point variation of less than 2% assures the homogeneity of the surface degradation.”

  1. Line 280: Figure 1A label to be changed form A-C to C-A.

We thank the reviewer; we changed the label as suggested

  1. Line 285: Table 1: Why does the S-300°C-1h has high luminosity of 92.2? It is the highest reported values for S films.

For what concern the reflectance measurements we have average information about the surface, and the colorimetric coordinates can be influenced by the grain dimensions. In addition, the formation of  new compounds (CdO and CdSO4) plays a key role. If the temperature increases the grain size of the sample is higher, and this means a decreasing in luminosity but also a major formation of Cadmium sulfate and a presence of a darker compound as the CdO. So, the effect of the co-presence of whitish and a darker compound, in addition to the variation of grain sizes of the sample, can influence the variability of the L parameter, which does not follow a linear increase as a function of the heat treatment. We used this technique only to get an indication of possible degradation which is confirmed and identified by other analytical techniques. To take into account this topic, we rephrased the complete paragraph (line 312-322)

  1. Line 288: Did authors observe similar shift in S-A samples after calcinations of various times at 500°C? supplement the data.

      We thank the reviewer for the comment. The shift of S-sample is not  remarkable  as can be seen from the Xo values  in the table 1, the peaks of the first derivative ranging from  514.5 and 556 (for the no-treated S-sample)  to 515 and 556 (for S-500-6h). We added this sentence in the text (line 303)

  1. Line 138-140: “It was decided”. Could authors share an explanation or cite a reference for relation between size of the NPs to oxidation rate/ ionic silver release.

We thank the reviewer for the comment, but, if we correctly understood his/her question, we could not answer because we did not treat the oxygen reactivity of CdS NPs as a function of grain size in our work. Although it is an interesting topic, partially treated by Comelli et al, we did not perform a study in presence of CdS NPs and as a function of their size grain.  

All the changes in the text are reported in yellow

Reviewer 3 Report

This topic is of interest and systematic studies looking at defects and structural changes within CdS are important. The title of the paper doesn’t mention XRD/SEM-EDS but the abstract and conclusion both discuss the value of XRD to comment on formation of compounds. However, the results don’t really support these statements. It’s also unclear what real value Raman has added to knowledge of degradation product formation, especially in the context of artificial aging at high T/long times. If there are superficial clusters present, which aren’t supported by the Raman / XRD results, would SEM-EDS cross-sectional imaging not show this? I don’t think the arguments for superficial clusters are well supported here.

The narrative overall is a bit hard to follow. Because there is a lot of data and plots (clearly the authors have done a lot of work!), I wonder if it might be best to present the data and then make a separate discussion section synthesizing the information from all of the techniques. It might also be useful to comment on whether we are at a disadvantage with the artificial aging process in these studies.

The beginning of the introduction needs citations. Additionally, the first four references discussing techniques are self-citations. Pump probe spectroscopy has indeed been applied by other researchers in CH (i.e. Warren Warren and Martin Fischer).

Not all issues in English language are listed in the attached suggested edits. A thorough reading by an English language expert is recommended.

Author Response

We thank the reviewer for all the comments. We answered point by point. All the changes in the text are evidenced in yellow.

This topic is of interest and systematic studies looking at defects and structural changes within CdS are important. The title of the paper doesn’t mention XRD/SEM-EDS but the abstract and conclusion both discuss the value of XRD to comment on formation of compounds.  

We thank the reviewer for the comment, we change the title mentioning XRD (SEM/EDS is only in the supplementary)

However, the results don’t really support these statements. It’s also unclear what real value Raman has added to knowledge of degradation product formation, especially in the context of artificial aging at high T/long times. If there are superficial clusters present, which aren’t supported by the Raman / XRD results, would SEM-EDS cross-sectional imaging not show this? I don’t think the arguments for superficial clusters are well supported here.

We thank the reviewer for her/his comment. We answer by proposing the following considerations:

  • The complexity of Cd pigments degradation is studied in literature and the formation of degradation products is known to be related to the contemporary presence of CdO and CdSO4. However, their formation mechanisms, the reactivity to the environment, and the role of some precursor elements (like point defects) are still not completely understood.
  • As evidenced in the text, the role of Raman spectroscopy is fundamental to detect the presence of sulfate compounds formed as a thin patina or clusters on the surface. The band around 1000 cm-1 is characteristic for this species and it is strong enough to study it in relation to the temperature and with respect to the similar band from the barium sulfate, also present on C
  • The relative content of sulfates presents in this patina, with respect to the overall sample, cannot be revealed by XRD until high temperatures (500°C). On the contrarty, Raman spectroscopy detected this compound also at lower temperature as reported in the sentence “The existence of this band is revealed also in the samples treated at lower temperature than 500°C.” (line 398).
  • From literature, the presence of sulfates on the surface cannot exclude the contemporary formation of CdO, not detectable with Raman spectroscopy. SEM-EDS on the surface or in cross-sectional mode does not assure the presence of CdO because it cannot distinguish between oxygen in cadmium sulfate or cadmium oxide, both compounds being contemporary present. We only obtain this result by operating a stoichiometric calculation of the oxygen in saturation of S and Cd, as done in the text (see for example supplementary materials and table 4 in the text)
  • So, the role of Raman spectroscopy is well identified as the non-destructive technique sensible enough to detect the surface composition of at least of one of the possible compounds due to the degradation process.
  • With the help of other non-destructive techniques, we confirmed the presence of sulfates, through colorimetric parameters, and we tried to determine the presence of CdO by studying the precursor elements that can origin its formation. For example, through TRPL and pump probe measurements we confirmed the presence of S vacancies, precursor element to CdO formation.
  • In particular with pump probe absorption spectroscopy, we tried to study the role of point defects that can be considered the precursors to the formation of sulfates easily detected by Raman spectroscopy.

The narrative overall is a bit hard to follow. Because there is a lot of data and plots (clearly the authors have done a lot of work!), I wonder if it might be best to present the data and then make a separate discussion section synthesizing the information from all of the techniques. It might also be useful to comment on whether we are at a disadvantage with the artificial aging process in these studies.

We thank the reviewer for the comment, and we appreciate the recognition of our work. Since numerous techniques are used in this study, and the complexity of the topic in relation to the superficial effect of degradation, we analysed point by point the information provided by single technique. We preferred building our line of reasoning by adding step-by-step each information in order to confirm or evidence the problematic interpretation. In our opinion, after numerous approaches of the paper writing, to avoid repetitions, we concluded that this way is the most reasonable. 

The beginning of the introduction needs citations. Additionally, the first four references discussing techniques are self-citations. Pump probe spectroscopy has indeed been applied by other researchers in CH (i.e. Warren Warren and Martin Fischer). Not all issues in English language are listed in the attached suggested edits. A thorough reading by an English language expert is recommended.

We added references as suggested, in particular Fischer and Warren works were included.

We thank the referee for his/her comment, and we made changes in the text based on his suggestions. We have also added references.

Line

  • 17 – ``spectroscopy’’

We corrected on the text.

  • 31 – “most famous pigments” – this seems to be an overstatement and not cited for Support
  • The first two paragraphs need citation. Additionally, there needs to be citations of other research groups doing pump probe spectroscopy in CH, which do exist. The authors have only cited really themselves.

We added new citations as required

  • 62 – use Greek alpha symbol instead of alfa changed
  • 120 – paint’s or paints’ changed
  • 131 – un-bold “by” changed
  • 141 – need to introduce orange CdS – only yellow was discussed

We introduced it in the introduction section, adding this paragraph.

“During the 20° century, to produce different hues from the light yellow typical of CdS, its synthesis started to be changed by inserting zinc to lighten the color, and selenium to increase the red hue. The use of selenium during the production allowing to obtain a range of colors from pale yellow to red, depending on the percentage of selenium, and these pigments are known as cadmium sulfoselenide compounds. Around the 1920, to obtain another lighter shade, the use of additive barium sulfate was included in the synthesis procedure.  These pigments can be prepared by calcinating again the hexagonal CdS plus selenium, or calcinating the precipitate resulting from the treatment of a cadmium salt with an alkaline selenide and sulfide[3].

    For what concern the degradation studies cited in the introduction related to yellow pigments, they have validity also for the orange hues. 

  • 274 – “show” Corrected  
  • 281 - The figure labeling of figure 1 is confusing – A, B don’t stick with the plots. Also, it’s not consistent with the following figures notation.                                          

We corrected it

  • 281 – why is the 1 st derivative of SA not shown?

    We did not show it because after the treatments the variation on its first derivative was not remarkable as in the case of the C-samples. We put only the values of Xo in table 1 to indicate the very small variation induced on these samples (less than 1 nm as variation). We introduced an explanation sentence in the text

  • 333 – how is the reduced tau2 time for C-500 6h explained? This was not clear.

The formation of Cadmium oxides and cadmium sulphate are both present in the reaction taking place at 500°C  as demonstrated in [4].  We may argue that, by leaving the sample for several hours at the same temperature (500°C ), we are favouring the formation of one compound (the sulfate) over the other (the oxide), thus causing a reduction of tau 2 , which, for large value, can be attributable to the oxide species, as written on the paper.   

  • 372 – a word is missing after revealed? changed
  • 416 – figure 4c – the deltaE plot showing the exponential curve seems rather force. Also, according to Equation 1, what is lambda?    

We apologize for the mistake, the letter lambda is an error, and the value of the numerator in the exponential is the exposure time (expressed in hours). We thank the referee for pointing out the error. We included the correction in the text.    The corrected equation is proposed in the text.

With this fit, we have only tried to follow a possible trend of the total induced variation. The data, delta E, distributed in this way only allows indicative fits to be made.           

  • 474 – figure 5 – these plots need titles for easier understanding of what each plot is

 We inserted a detailed label for the figure 5 in the text.

  • 516 – figure 6 – units/axes/lables needed. Again, it seems plot labeling is inconsistent throughout the paper, the previous figure didn’t utilize a/b/c…

We modified the figure accordingly.

  • 547 – I don’t think this is a robust/correct summary. As the authors wrote in 343, Raman cannot make a statement on CdO formation and the XRD results (as discussed in 268) did not show CdO or CdSO4 below 500 C, only some peak broadening. That conclusion should be reworded, and I’m not sure there is a clear story for what Raman/XRD is showing us about defects and structural changes. The abstract of the paper, in line 15, also seems to overstate what these techniques are concretely showing us. I think this needs to be re-thought, and perhaps other techniques need to be suggested to address the structural analysis and small changes. Or maybe the experiment itself needs to be changed to see the changes better.

We thank the reviewer for the comment. We already answered in detail to this topic before. We inserted some changes in the conclusion section to make it clearer.

For what concern the formation of CdSO4 and CdO, we detected both of them in the XRD pattern of 500°C at 6h, before with this technique, the amount of the degradation products is indetectable in the XRD, only a broadening was present. Therefore, the formation of CdO was confirmed by XRD and also by SEM-EDX analysis in both the S-500-6h and C-500-6h samples.   

Whilst sulfate compounds are detected by Raman spectroscopy, obtaining proof of the presence of CdO with this technique is very difficult because, as it is written in the text, CdO is not Raman-active, so we can’t obtain peaks for identification.  

Generally, Raman spectroscopy can be used to identify phase changes or the formation of new alteration products, as reported in several works in literature, but for some crystal phases, as in the case of CdO, it is not possible to prove its formation through Raman spectroscopy. For this reason, we have combined other qualitative techniques to achieve the definitive identification of this degradation product.  

Furthermore, we performed also TRPL, in the ps-scale, gathering info about changes in the decay times. Those changes agree with the ones reported in the literature for CdO, whose formation is also supported by the reflectance changes on the sample. Finally, the formation of defects due to the thermally driven formation of new compounds is also confirmed by Transient absorption and time-resolved luminescence. All these indirect indications corroborate the data provided by Raman and XRD measurements.